# A Specific Emitter Identification System Design for Crossing Signal Modes in the Air Traffic Control Radar Beacon System and Wireless Devices

**DOI:** 10.3390/s23208576

**Published:** 2023-10-19

**Authors:** Miyi Zeng, Yue Yao, Hong Liu, Youzhang Hu, Hongyu Yang

**Affiliations:** 1School of Computer Science, Sichuan University, Chengdu 610065, China; mickyzen@foxmail.com (M.Z.); liuhong@scu.edu.cn (H.L.); 2Sichuan Jiuzhou Electric Group Co., Ltd., Mianyang 621000, China; yueyao@my.swjtu.edu.cn (Y.Y.); huyouzhangyl@163.com (Y.H.)

**Keywords:** specific emitter identification (SEI), radio frequency fingerprint (RFF), multiple modal, unify features, air traffic control radar beacon system (ATCRBS)

## Abstract

To improve communication stability, more wireless devices transmit multi-modal signals while operating. The term ‘modal’ refers to signal waveforms or signal types. This poses challenges to traditional specific emitter identification (SEI) systems, e.g., unknown modal signals require extra open-set mode identification; different modes require different radio frequency fingerprint (RFF) extractors and SEI classifiers; and it is hard to collect and label all signals. To address these issues, we propose an enhanced SEI system consisting of a universal RFF extractor, denoted as multiple synchrosqueezed wavelet transformation of energy unified (MSWTEu), and a new generative adversarial network for feature transferring (FTGAN). MSWTEu extracts uniform RFF features for different modal signals, FTGAN transfers different modal features to a recognized distribution in an unsupervised manner, and a novel training strategy is proposed to achieve emitter identification across multi-modal signals using a single clustering method. To evaluate the system, we built a hybrid dataset, which consists of multi-modal signals transmitted by various emitters, and built a complete civil air traffic control radar beacon system (ATCRBS) dataset for airplanes. The experiments show that our enhanced SEI system can resolve the SEI problems associated with crossing signal modes. It directly achieves 86% accuracy in cross-modal emitter identification using an unsupervised classifier, and simultaneously obtains 99% accuracy in open-set recognition of signal mode.

## 1. Introduction

With the great advancements in science and technology, wireless devices have been applied in various fields, such as radar systems, the Internet of Things (IoT), and more. As a result, the security of wireless communication has attracted considerable attention [1]. The address resolution of the transmitted protocol is a traditional method used to identify wireless devices; however, it has practical limitations, as it can be easily intercepted and tampered with. To resolve these problems, reference [2] first proposed the concept of a radio frequency fingerprint (RFF) for transient signals, which is unique to each wireless transmitter. The various manufacturing biases in different hardware components of wireless devices result in each device having a unique RFF, like human fingerprints. Moreover, reference [3] explored a method to extract RFF from steady signals, which are easier to collect. This further advanced the development of RFF technology.

The enhanced RFF technique greatly improves the security of identification within the wireless domain. Furthermore, the introduction of the RF machine learning system by the Defense Advanced Research Projects Agency (DARPA) has propelled RFF research to pivot toward deep learning (DL) [4]. With the improvement of RFF and DL [5,6], SEI has also made great progress in distinguishing specific emitters based on physical layer characteristics. Typically, popular SEI systems consist of an RFF feature extractor and a DL classifier. References [7,8,9,10] extracted robust RFF features to improve the performance of SEI in complex environments. Moreover, references [11,12] proposed complex-valued convolutional neural networks (CVCNNs) to enhance the adaptability of the DL model when dealing with complex-valued signals in SEI, and compressed CVCNNs to save calculation resources.

Although there have been many breakthroughs in the field of SEI, previous works have mainly focused on optimizing a typical signal for higher robustness. In practical applications, working devices tend to use more than one modal signal to address the vulnerability or interference of a single modal signal in challenging environments. Multimodal signals usually operate in different frequency bands or have different modulation methods, which can effectively solve this problem. For example, in fast flight mode, high-frequency signals are extremely sensitive to Doppler shifts, while low-frequency signals are less affected by speed changes. Moreover, aircraft broadcast multi-modal signals, such as automatic dependent surveillance–broadcast (ads-b) signals, air traffic control (atc) signals [13], and radar signals, enabling accurate detection and guidance by air traffic controllers on the ground. Reference [14] explored the RFFs of aircraft based on radar signals. Moreover, references [15,16,17] explored the RFFs of airplanes, targeting the ads-b s modal signal. The traditional ads-b signal represents the ads-b s mode, which is also used in this paper. We use rm1/2 to represent the ads-b a/c mode in this paper, as seen in Table 1. Moreover, civil ATCRBS contains the ads-b a/c/s mode, which might be emitted by different parts of the same aircraft. Therefore, it is crucial for a robust SEI system to recognize wireless devices across different modal signals.

Typically, existing SEI systems [5,6,10,19,20] consist of RFF feature extractors and emitter classifiers, both of which are based on a single modal signal. However, when dealing with multi-modal signals, it is necessary to first identify the signal mode as a signal arrives. Moreover, different modal signals require different RFF extractors and SEI classifiers, resulting in increased complexity and reduced robustness when dealing with unknown modal signals. To tackle the challenge of identifying wireless devices across different modal signals in a more cost-effective manner, one possible solution is to develop a universal RFF extractor that unifies the structures and semantics for multi-modalities. Additionally, it is crucial to introduce DL methods to improve the adaptability of classifiers for multi-modal signals [21,22]. A previous study [21] proposed pre-training a SEI DL classifier using ads-b signals and subsequently fine-tuning it to recognize emitters with Wi-Fi signals. However, this approach requires continuous fine-tuning whenever new modal signals are introduced, which also leads to building multiple models for different modal signals.

To resolve these issues, the MSWTEu is proposed to extract general and robust RFF features, unifying the structures and semantics of different modal signals. However, solely relying on the general RFF extraction method may not be sufficient, as some modal signals exhibit significant differences that MSWTEu cannot fully unify. To address this limitation, we propose a novel FTGAN to enhance the adaptation of the entire SEI system to multi-modal signals. Since it is impractical to build signal models for all existing modal signals, we focus on analyzing and obtaining information from one representative modal signal, which we refer to as the “known modal signal“. The remainder of the modal signals are deemed as “unknown modal signals”. The FTGAN tries to transfer the MSWTEu distributions of unknown modal signals to match the distribution of the known modal signal. With the above methods, only one SEI classifier is required to be built and trained for the known modal signal. But when confronted with other modal signals, the SEI classifier can also recognize their emitters and achieve improved comprehensive accuracy recognition. Furthermore, with slight modifications to the trained SEI classifier, open-set signal mode recognition can be achieved. An overview of traditional SEI systems, as well as our system addressing multi-modal signals, is depicted in Figure 1.

The main contributions of this paper are as follows:1.We explore an efficient and general method for extracting prominent RFF features from multi-modal signals;2.We explore the MSWTEu algorithm for unifying the structure and semantics of different modal signals;3.We explore an FTGAN to further unify RFF features, thus enhancing the efficiency and performance of the SEI system while accommodating multi-modal signals. With this approach, the SEI classifier is trained only with known modal signal features, but it can also recognize emitters using unknown modal signals without additional training;4.We leverage MSWTEu and FTGAN; the trained SEI classifier can achieve open-set modulation recognition with minimal fine-tuning;5.We construct a hybrid dataset for SEI experiments crossing signal modes, consisting of a real dataset, a simulated dataset, and a public dataset. The public dataset has been widely used in recent works and consists of Wi-Fi signals. The simulated dataset comprises two modal communication reply signals, whose waveforms are carefully designed and generated by different waveform generators to simulate emitters. The real dataset is comprised of ads-b signals collected over a span of 12 months; it has undergone comprehensive preprocessing. Each modal signal is sent by three different emitters. The details of the dataset are depicted in Table 1. The simulated dataset and real dataset consist of ads-b a/c/s modal signals, which can be built as a complete dataset of civil ATCRBS (CATCRBS). With the datasets, our optimized SEI system can obtain the identification of airplanes, disregarding variations in signal modes and, thus, replacing the identification-decoding module within CATCRBS.

Therefore, our proposed system can solve the SEI problems posed by multi-modal signals at the same time: (1) MSWTEu allows different modal signals to only utilize one RFF extractor, decreasing the number of RFF extractors; (2) FTGAN ensures that different modal signals require only one unsupervised classifier, solving the hard labeling work and reducing the number of classifiers; (3) the proposed training strategy for MSWTEu and FTGAN makes it unnecessary to add additional processes to recognize signal modes, increasing the efficiency of the whole system; (4) a little fine-tuning of the proposed SEI system enables open-set signal mode recognition. The rest of the paper is organized as follows. Section 2 describes the process of the feature extractor MSWTEu. In Section 3, we present the details of FTGAN and the training strategy. The dataset details and experimental results are shown in Section 4. Lastly, Section 5 summarizes this paper.

## 2. RFF Feature Extracting and Unifying MSWTEu

This work is mainly inspired by our previous work [13], which proposed a robust RFF extractor denoted as improving SWT by energy regularization (ISWTE), enhancing the performance of SEI at low SNRs. In a previous work [13], we fully discussed the advantages of wavelet transformation in improving RFF features and conducted experiments to compare ISWTE, SWT, spectrograms, and traditional signal processing methods. We concluded that SWT outperforms other RFF extractors in terms of accuracy and efficiency. Moreover, by optimizing SWT with energy regularization, ISWTE can enhance the performance of SEI under low SNRs. However, this kind of enhancement is limited to one particular modal signal. Thus, our aim is to expand our previous work, and further extract general and robust RFF features from multi-modal signals.

This work is roughly divided into three stages, as shown in Figure 2.

In the first stage, we propose a reasonable downsampling strategy to unify the lengths of multi-modal signals. This strategy helps to address the signal variation issue and ensures the effectiveness of subsequent processing.

The second stage focuses on improving the versatility and efficiency of the MSWTEu method based on the ISWTE method. While ISWTE decomposes signals into a fixed, limited number of layers, and adopts fixed parameters, which are manually adjusted to achieve the best performance, MSWTEu allows for optional layers and optional parameters for each layer. We propose the MSWTEu, which introduces an optional parameter for each layer, and design an adaptive selection algorithm to select the best SWT decomposition layers and the best parameter per layer. Here, we use the known modal dataset and pick a few unknown modal signals to join the selecting algorithm. The accuracy of a clustering algorithm is adopted to assess the performance. And in turn, the layers and parameters are adaptively determined by the best performance. By doing so, the MSWTEu method combines the advantages of multiple SWT layers and the automatic selection of the best parameters, eliminating the need for manual adjustments. This approach ensures that different modal signals have a unified structure and shape, and also achieves robust RFF feature extraction.

In the third stage, we address the energy explosion issue commonly encountered in SWT transformation and optimize MSWTEu for greater efficiency. The energy explosion is mainly concentrated in the latter decomposition layer from SWT, resulting in a substantial increase in the weight of these latter layers in the RFF feature. It could cause the information of previous layers to be missing. Therefore, the ISWTE is optimized by energy regularization, specifically, by computing the energy of each layer and reversely weakening them by an energy scale. In this way, ISWTE has achieved good performance in noisy domains based on ads-b signals. However, the time complexity of energy regularization by ISWTE is not ideal since the procedure of reverse weakening is complex. The MSWTEu aims to simplify the energy strategy and be more adaptive to multi-modal signals.

In conclusion, the MSWTEu overcomes the constraints of relying on one modal signal and manual parameter tuning during RFF feature extraction. It improves the efficiency of energy computation and resolves the problem of energy explosion.

The MSWTEu calculation process is shown in Figure 2 and the details are below:

(1) **Wavelet transformation (WT)**: In practice, signals are usually downsampled upon reception and transformed to the discrete domain. Therefore, the signal is represented as x(n), not x(t). The normal wavelet transformation (WT) is calculated as below: (1)$$W_{a,b}\left(x\left(n\right)\right)\;=\;\frac{1}{\sqrt{a}}\sum_{n=0}^{N}x\left(n\right)\varphi \left(\frac{n-b}{a}\right),$$
where *N* is the length of x(n), φ is the mother wavelet function, *a* scales the wavelet function, and *b* shifts the time domain. The WT can transform signals from the time domain to the time–frequency domain, by slicing the frequency domain based on the continuous short-time Fourier transform (STFT). Reference [3] proposed that the signal in the frequency domain performs better in SEI than in the time domain. Our previous experiments show that the signals in the time domain are vulnerable to noises but robust in Doppler shifts, while the signals in the frequency domain behave in the opposite way. The WT combined the advantages of the time domain and frequency domain, which is the main reason we adopted it as the base transformation of the RFF extractor. Normally, the WT can decompose signals into multiple layers, with each containing a high-frequency part (the detail coefficient) and a low-frequency part (the approximation coefficient).

(2) **Synchrosqueezed wavelet transformation (SWT)**: Reference [23] proposed SWT, which is inspired by empirical mode decomposition (EMD) and confirms its effectiveness through mathematical methods. Unlike the traditional discrete wavelet transform (DWT), EMD does not require much prior knowledge and is more stable and adaptive in signal decomposition and feature extraction. Moreover, DWT reduces the length by half with each decomposition to the subsequent layer, whereas SWT maintains the same length at each layer, making it challenging to evaluate the length and unify the shapes of multi-modal signal features. Therefore, we adopt SWT as the main transformation for signals. SWT can be calculated as below: (2)$$W_{s\;\;}\left(a,b\right)=\;A/4\pi \;\cdot a^{1/2}\cdot \varphi \left(a\omega \right)e^{nb\omega },$$
(3)$$W_{a,b}^{\prime }\left(x\left(n\right)\right)\;=\;\sum_{n=0}^{N}x\left(n\right)W_{s\;\;},$$
(4)$$\begin{array}{cc} \; & {SWT}_{1}\left(x\left(n\right)\right)=\\ \; & W_{a,b}^{\prime }\left(x\left(n\right)\right)*L\left(n\right),W_{a,b}^{\prime }\left(x\left(n\right)\right)*H\left(n\right)\;\;n\leq N,\\ \; & {SWT}_{2}\left(x\left(n\right)\right)=\\ \; & {W_{a,b}^{\prime }(SWT}_{1g})*L\left(n\right),{W_{a,b}^{\prime }(SWT}_{1g})*H\left(n\right)\;\;\;\;n\leq N,\\ \; & \ldots \ldots , \end{array}$$
where ${SWT}_{1g}=W_{a,b}^{\prime }\left(x\left(n\right)\right)*L\left(n\right),\;{SWT}_{2g}={W_{a,b}^{\prime }(SWT}_{1g})*L\left(n\right)\ldots \ldots ,\;$$W_{s}\left(a,b\right)$ is given and proved by reference [23], L(n) is a low-pass filter, and H(n) is a high pass filter. With L(n) and H(n), SWT can obtain detailed and approximate information for each layer. In Figure 2, ${SWT}_{1h}=W_{a,b}^{\prime }\left(x\left(n\right)\right)*H\left(n\right),\;{SWT}_{2h}=W_{a,b}^{\prime }\left(SWT_{1g}\right)*H\left(n\right)$.

(3) **Multiple synchrosqueezed wavelet transformations (MSWT)**: Utilizing SWT based on SFST, signals are sliced from long-term time domain features into multiple short-frequency domain features, transforming from the time domain to the time–frequency domain. Therefore, SWT can obtain more obvious RFF features and extract multiple-layer features from one signal. Each layer feature includes an approximation feature and a detail feature. However, during experiments, the RFF features of different modal signals exhibit varying performance in each layer, which poses a challenge in selecting the appropriate and general decomposition layer for multi-modal signals. One work [24] proposed the method of downsampling one dataset into several subsets to replace one single dataset, augmenting the datasets [25] and enhancing the pre-existing knowledge. However, this approach has limited impact on the improvement of the SEI system and requires significant computational resources for training the classifier. Based on the idea of multiple data enhancement, MSWT is proposed to integrate the details and approximations information from multiple SWT layers, effectively enhancing the RFF feature extraction and disregarding the disparate performances of different modal signals across various SWT layers. With it, feature extraction is simple and easy, but with strong and general features.
(5)$$\begin{array}{cc} \; & MSWT=M\cdot [{SWT}_{1}\left(x\left(n\right)\right).T,{SWT}_{2}\left(x\left(n\right)\right).T,\\ \; & \ldots ,{SWT}_{L}\left(x\left(n\right)\right).T].T\;\;\;M\in R^{1\times L}\\ \; & \to \\ \; & MSWT=[m_{1g},m_{1h},m_{2g},m_{2h},\ldots ,m_{Ng},m_{Nh}]\cdot \\ \; & [SWT_{1g},SWT_{1h},\ldots ,SWT_{Ng},SWT_{Nh}].T, \end{array}$$
where *N* is the max decomposition layer of SWT for limitation, which is the same as the signal length, and m∈[0,1] represents the weight of each decomposition layer using *g* and *h*.

(4) **RFF structure unifying**: The MSWT is still a complex-valued feature with a real part and an imaginary part, making it challenging to unify the structures of different modal features. In this paper, we focus more on unifying the structures of different modal signals for SEI while maintaining the effectiveness of each modal RFF feature. To accomplish this, we incorporate FFT to further extract RFF features for enhanced frequency information [3] and apply a modulus operation to unify the structures. The MSWT can be redefined as follows:(6)$$\begin{array}{c} {\mathrm{MSWT}}_{m}=\sqrt[{2}]{{re(fft(MSWT))}^{2}+{im(fft(MSWT))}^{2}}, \end{array}$$
where fft is the fast Fourier transform, re() and im() are the real and imaginary parts of the complex-valued feature.

(5) **Special particle swarm optimization (SPSO)**: $MSWT_{m}$ has successfully addressed the challenges of unifying and enhancing the effectiveness of RFF features from multiple modalities. However, the problem of determining the appropriate number and weights of layers remains unresolved due to the inefficiency of manual adjustment. In order to minimize parameter adjustment efforts and optimize the effectiveness of RFF features, it is necessary to implement an adaptive algorithm in $MSWT_{m}$ to determine and unify the best combination of layers and their respective weights for multi-modal signal features. This algorithm can adaptively achieve high RFF performance while using fewer layers and be applicable to all modal signals. To address this issue, we propose an SPSO, which is an adaptive selection algorithm for $MSWT_{m}$ based on PSO [26]. As in Figure 2, the best $MSWT_{m}$ requires appropriate layer decomposition and proper weights for each layer, including *g* and *h*. The SPSO selects these parameters; the details are as follows. (This paper does not seek to optimize the traditional selection algorithm, PSO, but only applies it to suit our RFF extractor).

1:Our dataset has four modal signals and each mode has three devices. Each device introduces 20 labeled signals to join the SPSO selecting procedure. The details of the dataset are presented in Table 1;2:The initial *l* is set to 1 and SPSO reaches the best clustering results by selecting $\left\{m_{1g},m_{1h}\right\}$, $\;0\leq m_{1g}\leq 1,0\leq m_{1h}\leq 1$. The features are clustered for different emitters, and the performance is evaluated by the mean distance between each point and its clustering center. The minimum mean distance is denoted as $ds_{1}$, when l=1;3:l=l+1, and SPSO selects $\left\{m_{1g},m_{1h},\ldots ,m_{lg},m_{lh}\right\},\;0\leq m_{ig}\leq 1,0\leq m_{ih}\leq 1$, as above. The minimum mean distance is denoted as $ds_{l}$;4:Repeat step 3 until $ds_{l}>\left(ds_{l-1}+ds_{l-2}+ds_{l-3}\right)/3\;if\;l>3$, selecting l=l−3;ifl<3orl=3 (some modal signals cannot be decomposed to more than 3 layers); select *l* when obtaining the minimum $ds_{l}$. The mean distance is calculated as follows:
(7)$$ds=(\sum_{t=1}^{r}\sqrt[{2}]{\sum_{j=1}^{f}{\left(p_{tj}-c_{kj}\right)}^{2}})/r,$$
where *r* is the total signal number in clustering, *f* is the unifying feature length, and $c_{k}$ is the clustering center for $p_{t}$.

(6) **Energy balancing and multiple synchrosqueezed wavelet transformation of energy unified (MSWTEu)**: Combining the above steps, the RFF features of different modal signals can be unified into a unified and efficient structure. However, the unification of different modal RFF features is still inadequate, as the inherent properties and distributions are not fully unified. Moreover, the SWT energy explosion issue remains unresolved. Reference [13] proposed that energy regularization for combining SWT layers can enhance the RFF features. Furthermore, an energy-balancing strategy can unify the semantics of different modal signals and improve RFF, as energy accumulation and computation have always been important for signal processing and analysis in emitter identification. We expand upon our previous work [13] to accommodate multi-modal signals and simplify the energy-balancing algorithm, resulting in increased recognition speed. Subsequent experiments have compared the complexity and time cost between ISWTE [13] and MSWTEu. With energy balancing, the performance of MSWTEu also exhibits significant improvements in identification, as depicted in the following experiments. The implementation details of energy balancing by MSWTEu are as follows.

The total energy and average energy can be calculated:
(8a)$$\begin{array}{cc} \; & g_{i}\left(w\right)=SWT_{ig}^{\prime }\left(w\right)=fft\left(m_{ig}SWT_{ig}\left(n\right)\right), \end{array}$$
(8b)$$\begin{array}{cc} \; & h_{i}\left(w\right)=SWT_{ih}^{\prime }\left(w\right)=fft\left(m_{ih}SWT_{ih}\left(n\right)\right), \end{array}$$
(9a)$$\begin{array}{cc} \; & g_{i}^{\prime }\left(w\right)=SWTE_{ig}^{\prime }\left(w\right)=\frac{\sum_{w=1}^{w=N}g_{i}^{2}\left(w\right)}{\sum_{j=1}^{j=l}\sum_{w=1}^{w=N}g_{j}^{2}\left(w\right)}, \end{array}$$
(9b)$$\begin{array}{cc} \; & h_{i}^{\prime }\left(w\right)=SWTE_{ih}^{\prime }\left(w\right)=\frac{\sum_{w=1}^{w=N}h_{i}^{2}\left(w\right)}{\sum_{j=1}^{j=l}\sum_{w=1}^{w=N}h_{j}^{2}\left(w\right)}, \end{array}$$
(10a)$$\begin{array}{cc} \; & E_{totalg}=\sum_{j=1}^{j=l}\sum_{w=1}^{w=N}g_{j}^{2}\left(w\right), \end{array}$$
(10b)$$\begin{array}{cc} \; & E_{totalh}=\sum_{j=1}^{j=l}\sum_{w=1}^{w=N}h_{j}^{2}\left(w\right), \end{array}$$
(11a)$$\begin{array}{cc} \; & E_{aveg}=\sum_{j=1}^{j=l}\sum_{w=1}^{w=N}g_{j}^{2}\left(w\right)/l, \end{array}$$
(11b)$$\begin{array}{cc} \; & E_{aveh}=\sum_{j=1}^{j=l}\sum_{w=1}^{w=N}h_{j}^{2}\left(w\right)/l, \end{array}$$
where $g_{i}$ and $h_{i}$ represent the detailed and approximation information of each decomposition layer, respectively, and are unified in the structure according to (6). On the other hand, $g_{i}^{\prime }$ and $h_{i}^{\prime }$ are the relative weight values, which represent the ith weight value among the total *g* and *h* values across the *l* layers. By integrating these relative values into MSWTE and substituting MSWT with them, we can effectively attain the targeted result: (12)$$MSWTE=[g_{1}^{\prime },h_{1}^{\prime }\ldots g_{l}^{\prime },h_{l}^{\prime }].T.$$

Based on MSWTE, MSWTEu is designed to unify and balance the energy of each layer. MSWTE is proposed to calculate the energy of each layer to a relative value, which effectively resolves the energy explosion problem. Moreover, MSWTEu proposes energy normalization based on the thought of normalization, which not only reduces the difference value between the maximum and minimum energies but also ensures a consistent energy across each layer.
(13)$$\begin{array}{cc} \; & MSWTEu=[g_{1}^{\prime }-{\frac{E_{aveg}}{E_{totalg}}}_{2},h_{1}^{\prime }-{\frac{E_{aveh}}{E_{totalh}}}_{2}\\ \; & ,\ldots ,g_{l}^{\prime }-{\frac{E_{aveg}}{E_{totalg}}}_{2},h_{l}^{\prime }-{\frac{E_{aveh}}{E_{totalh}}}_{2}].T. \end{array}$$

Based on the above, when confronted with multi-modal signals, the MSWTEu demonstrates higher efficiency, better adaptivity, and fewer limitations compared to other RFF feature extractors [7,8,13,27], which have also been verified by subsequent experiments. Unlike these extractors, which are only based on one typical signal mode, requiring extensive manual tuning or having high complexity, the MSWTEu offers a unified approach to extracting universal RFF features from different signal modes. This ultimately enables the SEI system to identify emitters based on a wider range of modal signals without the need for complex RFF extractors and additional classifiers.

## 3. RFF Feature Transferring—FTGAN

As depicted in Figure 3, there are four signal modes, and each mode can be transmitted by different emitters. Our proposed method can accurately identify the signal types and emitter IDs of all modal signals using only one modal signal, the SEI classifier. Specifically, we train an SEI classifier for the known modal RFF features, denoted as Mode 1, which has label and signal information. Then we utilize the network of classifier, excluding the softmax layer, as an identifying feature extraction network for all modal RFF features. Although the unknown modal (Mode 2/3/4) RFF features miss prior knowledge of labels and signals’ information, features extracted by the classifier network can be directly clustered into different emitters. With some modifications, the clustering can also be grouped into different signal modes. Some references [27,28] have also used clustering for evaluating and testing SEI performance. However, it is insufficient at extracting identifying features for all modal signals relying only on the network of one single modal SEI classifier. Therefore, we propose FTGAN to transfer the distribution of RFF features from unknown modal signals to match the distribution of the known modal signal.

There are three steps to complete the final clustering based on the above ideas: training the Mode 1 SEI classifier to identify feature extraction; training FTGAN to transfer unknown modal RFF features to match the known modal RFF features; combining the identifying feature extraction network and FTGAN to obtain identifying features from all modal signals to cluster emitters and signal modes.

**Training the Mode 1 SEI classifier and obtaining the identifying feature extractor:** To develop an SEI classifier for identifying emitters based on known modal signals, we utilize a ResNet model comprising three ResNet blocks and two dense layers. The ResNet model exhibits superior performance in managing gradient explosion compared to traditional convolutional networks. Commonly used networks for feature extraction, such as AlexNet, VGG, or GoogLeNet, have numerous layers, which might be excessive given the low dimensionality of signal features. Therefore, we adopt a more efficient network with fewer layers to build the SEI classifier, which also demonstrates excellent performance in classifying. Moreover, based on the classifier, we utilize its network up to the point before the softmax layer as the identifying feature extractor.

**Training FTGAN and matching all modal RFF features:** GANs have become popular in graph generation, but their architectures are not well-suited for waveform direct generation. The FTGAN has been inspired and upgraded based on GANs [13,22,29]. Figure 4 depicts the architectures of a normal GAN and FTGAN. Unlike traditional GAN designs, the FTGAN generator focuses on learning only the differences between different modes of signals. This approach allows for simpler networks and is more suitable for waveform calculation. Moreover, since the differences between different modes of signals have a lower information entropy compared to complete signals, the FTGAN is proposed to generate the feature difference between the features of known modal signals and unknown modal signals. Removing the feature differences from unknown modal signal features can achieve features matching with the known modal, which improves the accuracy and efficiency of waveform feature generation. The training procedure of FTGAN is as depicted in Figure 4: (1) There are two kinds of inputs, the known mode Mode 1 features denoted as *x* and the unknown mode Mode 2/3/4 features denoted as $x^{\prime }$. It is important to note that no labels are required during the training process of FTGAN; (2) $x^{\prime }$ is fed into $FTGAN_{g}$, which outputs the generated differences $FTGAN_{g}\left(x^{\prime }\right)$; (3) The unknown modal features by removing the differences obtain $x^{\prime }-FTGAN_{g}\left(x^{\prime }\right)$ to match with the known modal features *x*; (4) The discriminator is improved by enhancing the inputs. Normally, traditional $GAN_{d}$ is to calculate the similarity between *x* and $GAN_{g}\left(x^{\prime }\right)$ to determine whether inputting samples are real or generated ‘real’. $GAN_{g}\left(x^{\prime }\right)$ is the generated ‘real’ samples by GAN’s generator. Then, the similarity is used to update and enhance the parameters of $GAN_{g}$. The FTGAN is proposed to input $\left(x,x^{\prime }-x\right)$ and $\left(x^{\prime }-FTGAN_{g}\left(x^{\prime }\right),FTGAN_{g}\left(x^{\prime }\right)\right)$ to push the discriminator and also utilize the enhanced discriminator to encourage the generator. These inputs can calculate the similarity between real samples and generated ‘real’ samples, as well as the similarity between real differences and generated ‘real’ differences; (5) Three loss functions are added to improve the performance of generators and enhance the realism of the generated features.

**The entire process for clustering emitters and signal modes:** With the above steps, we can easily recognize different emitters and signal modes with high accuracy by achieving robust features for all modal signals. The entire process can be summarized as follows: (1) the MSWTEu extracts RFF features and unifies the structures of features for all modal signals; (2) all modal RFF features transformed by FTGAN are in a distribution matching that of Mode 1, enabling the Mode 1 SEI classifier network to extract clear identifying features for all modal signals; (3) With the identifying features extracted by the trained Mode 1 classifier network, all modal signals can be clustered by the emitter identification or signal mode. The following are the FTGAN details of the description and handling. The entire process can also be seen in Algorithm 1.
**Algorithm 1** The entire process for identifying emitters.**Step1**: Train the Mode 1 SEI classifier and obtain the identifying feature extractor**Input**: Known Mode 1 sample MSWTEu features *x* and labels**Output**: Identifying feature extractor exmodel1: Building Mode 1 classifier, as depicted in Figure 32: Training Mode 1 classifier by *x* and labels, as depicted in Figure 33: Obtaining exmodel from the trained Mode 1 classifier, as depicted in Figure 3**Step2**: Train FTGAN**Input**: Known Mode 1 sample MSWTEu features, denoted as *x*, unknown Mode 2/3/4 sample MSWTEu features, denoted as x’**Output**: The “Mode 1” MSWTEu features $x_{-}^{\prime }$ transformed from Mode 2/3/4 MSWTEu features x’**for** the number of training iterations **do**    Sample minibatch of m **Mode 1 samples** $x_{1},\ldots ,x_{m}$ from $D_{b}$    Sample minibatch of m **Mode 2/3/4 samples** $x_{1}^{\prime },\ldots ,x_{m}^{\prime }$ from $D_{t}$    Calculate ${\mathrm{FTGAN}}_{g}\left(x^{\prime }\right)$ and $L_{{\mathrm{FTGAN}}_{d}}$ by *x*, $x^{\prime }$, $FTGAN_{g}$, and (20b)     Update ${\mathrm{FTGAN}}_{d}$ parameters    Calculate ${\mathrm{FTGAN}}_{g}\left(x^{\prime }\right)$ and $L_{{\mathrm{FTGAN}}_{g}}$ by *x*, $x^{\prime }$, $FTGAN_{g}$, and (20a)     Update ${\mathrm{FTGAN}}_{g}$ parametersendfor$x_{-}^{\prime }$ = $x^{\prime }$ − $FTGAN_{g}\left(x^{\prime }\right)$**Step3**: Identify Emitters**Input**: Mode 1/2/3/4 sample MSWTEu features $x_{mul}$, emitter number $N_{emitter}$**Output**: SEI classifier clmodel and emitter id1: Load exmodel2: Transferring multi-modal features to match Mode 1 features:    $x_{mul_{-}}=x_{mul}-FTGAN_{g}\left(x_{mul}\right)$3: Extracting identifying features: $x_{mul_{i}}^{\prime }=exmodel\left(x_{mul_{-}}\right)$4: Building clustering model clmodel as in Figure 35: Unsupervised training clmodel by $x_{mul_{i}}^{\prime }$,$N_{emitter}$6: Randomly selecting 10 samples from each cluster for labeling7: $id=clmodel(x_{mul_{i}}^{\prime })$

The unlabeled samples from Mode 1 belong to the base domain, with their MSWTEu feature distribution represented as $D_{b}$. The unlabeled samples from Modes 2/3/4 exist within the transferred domain, characterized by their MSWTEu feature distribution $D_{t}$. The FTGAN aims to train $D_{t}\to D_{b}$, thereby minimizing and maximizing the adversarial objective function $\min_{G}\max_{D}V\left(D,G\right)$: (14)$$V\left(D,G\right)=E_{x}[log\left(D\left(x\right)\right]+E_{x^{\prime }}\left[log\left(1-D\left(G\left(x^{\prime }\right)\right)\right)\right],$$
where $x\in D_{b},\;x^{\prime }\in D_{t}$, *D* is the discriminator function, and *G* is the generator function of FTGAN.

Normally, GAN’s [30] loss functions of D and G are calculated as follows:
(15a)$$\begin{array}{cc} \; & L_{d}=E_{x}L_{b}\left(Y_{0},D\left(x\right)\right)+E_{x^{\prime }}L_{b}\left(Y_{1},D\left(G\left(x^{\prime }\right)\right)\right), \end{array}$$
(15b)$$\begin{array}{cc} \; & L_{g}=E_{x}L_{b}\left(Y_{0},D\left(x\right)\right)+E_{x^{\prime }}L_{b}\left(Y_{1},D\left(G\left(x^{\prime }\right)\right)\right), \end{array}$$
where $L_{d}$ denotes ascending and $L_{g}$ denotes descending while training GANs; $Y_{0}$’s size is D(x), filling with 0, and $Y_{1}$’s size is D(G(x)), filling with 1; $L_{b}$ is the popular binary function for loss calculation. The improving details and loss functions of FTGAN are optimized as follows:

(1) **Upgrading $L_{d}$ and $L_{g}$**: As depicted in the previous part, FTGAN is enhanced to be more robust by improving the inputs. Therefore, $L_{d}$ and $L_{g}$ are both upgraded:
(16a)$$\begin{array}{cc} \; & L_{d^{\prime }}=E_{x}L_{b}\left(Y_{0},D\left(\left[\begin{array}{c} x^{\prime }-x\\ x \end{array}\right]\right)\right)+E_{x^{\prime }}L_{b}\left(Y_{1},D\left(\left[\begin{array}{c} G(x^{\prime })\\ x^{\prime }-G\left(x^{\prime }\right) \end{array}\right]\right)\right), \end{array}$$
(16b)$$\begin{array}{cc} \; & L_{g^{\prime }}=E_{x}L_{b}\left(Y_{0},D\left(\left[\begin{array}{c} x^{\prime }-x\\ x \end{array}\right]\right)\right)+E_{x^{\prime }}L_{b}\left(Y_{1},D\left(\left[\begin{array}{c} G(x^{\prime })\\ x^{\prime }-G\left(x^{\prime }\right) \end{array}\right]\right)\right), \end{array}$$
where *x* is the known modal RFF feature, $x^{\prime }$ is the unknown modal RFF feature, $x^{\prime }-x$ is the real RFF feature difference between the known mode and the unknown mode, $G(x^{\prime })$ is the generated feature difference, $x^{\prime }-G\left(x^{\prime }\right)$ is the generated known modal RFF feature, $x^{\prime }-x$ is the real difference; $FTGAN_{d}$ simultaneously calculates the similarity between $\left(x^{\prime }-x,x\right)$ and the real, and the similarity between $\left(G\left(x^{\prime }\right),x^{\prime }-G\left(x^{\prime }\right)\right)$ and the fake; conversely, $FTGAN_{g}$ attempts to generate ‘real’ samples that cannot be discriminated by $FTGAN_{d}$. By calculating losses on the combination of differences and samples, the *G* and *D* of FTGAN can be trained more robustly. The enhanced inputs could significantly improve the generation ability of ‘real’ differences by FTGAN, enabling the transfer of unknown modal features to known modal features.

(2) **Transfer loss $L_{transfer}$**: To ensure the accuracy of the generated differences between $D_{t}\;and\;D_{b}$, the $L_{transfer}$ is added and the FTGAN is trained by descending $L_{transfer}$:(17)$$L_{transfer}=E_{x^{\prime }}L_{mae}\left(G\left(x^{\prime }\right),x-x^{\prime }\right)+E_{x^{\prime }}L_{mae}\left(x^{\prime }-G\left(x^{\prime }\right),x\right),$$
where $L_{mae}$ is the mean absolute error (MAE) loss function, which performs better than the mean squared error (MSE) in the signal waveforms [13]. By decreasing the loss between the generated differences and real differences, and the loss between generated known modal features and real known modal features, $L_{transfer}$ allows *G* to generate more reliable ‘real’ differences.

(3) **Amplitude loss $L_{amplitude}$**: To generate a more appropriate difference, the amplitude of generating value should be limited:
(18a)$$\begin{array}{cc} \; & {Loss}_{nm}=\sum_{i=1}^{bm}Nm(G\left(x_{i}^{\prime }\right))/bm-\sum_{i=1}^{bm}Nm({x_{i}^{\prime }-x}_{i})/bm, \end{array}$$
(18b)$$\begin{array}{cc} \; & L_{amplitude}=max\left(Y_{0}{,Loss}_{nm}\right), \end{array}$$
where Nm() is the binary regularization function, $Y_{0}$’s size is as $Loss_{nm}$, filling with 0, and bm is the batch size. Moreover, (18a) and (18b) allow the generated differences $G(x^{\prime })$ to fit the amplitude distribution of real differences $x^{\prime }-x$.

(4) **Smooth transition loss $L_{tn}$**: To smooth the MSWTEu features of the transferred signals, the FTGAN optimized it and descended it while training: (19)$$L_{tn}\;=E_{x^{\prime }}L_{mae}\left(\sum_{i=1}^{bm}G(x_{i}^{\prime })/bm\;-\;G\left(z^{\prime }\right)\right),\;$$
where $x_{epoch}^{\prime }=\left(x_{1}^{\prime },x_{2}^{\prime },\ldots ,x_{bm}^{\prime }\right)$ is a sample set of one epoch of x’, and $z^{\prime }\in x_{epoch}^{\prime }$. The loss function is added to improve the distribution balance of different modal features.

(5) **The FTGAN’s total loss function**: Above all, combining the added three loss functions and improved *G* and *D* functions, the final loss functions of FTGAN are as follows:
(20a)$$\begin{array}{cc} \; & L_{{FTGAN}_{g}}=\alpha L_{g^{\prime }}+{\beta L}_{transfer}+L_{amplitude}+L_{tn}, \end{array}$$
(20b)$$\begin{array}{cc} \; & L_{{FTGAN}_{d}}=L_{d^{\prime }}, \end{array}$$
where α=−0.5, and β=10.

## 4. Experiments and Discussion

### 4.1. Data Acquisition and Experimental Setup

**Dataset description:** Four modal communication signals, ads-b, Wi-Fi, and replied signals rm1 and rm2, were utilized in the experiments. Each modal signal was transmitted by three emitters. Table 1 provides details about the data employed. Ads-b signals were collected from flying airplanes over a period of 12 months, and pre-processed by us to an SNR of 10 dB. The collector used was the Tektronix RSA6120B real-time spectrum analyzer. At a frequency of 1090 MHz, approximately 5 modal signals were collected from the air acquisition. We screened ads-b signals from them and labeled them using the airplane’s International Civil Aviation Organization (ICAO) address. ICAO serves as the only global identifier [15,16], and can be obtained by decoding through our Simulink codes. Wi-Fi signals were obtained from a public dataset. The use of ads-b and Wi-Fi signals has been observed in previous SEI references [15,16,21,27]. Communication reply signals—rm1 and rm2—are two modal communication signals that use pulse position modulation (PPM). Rm1 and rm2 are similar to ads-b and are used to reply to interrogating signals in air traffic control, which are the ads-b a mode and ads-b c mode, respectively. Traditionally, the ads-b signal is broadcasted and easily received. The Ads-b a/c mode carries little information, and does not carry the airplane’s ICAO address; it is sent at specific time points. Therefore, we cannot receive enough signals with the a/c mode to label them by their ICAO. The ads-b s mode, a mode, and c mode employ different data links and do not share the same signal transmitters in the same airplanes. Therefore, we can simulate the a/c mode by other emitters, which are called rm1 and rm2 in this paper. Based on ads-b, rm1, and rm2, we can completely simulate CATCRBS and conduct SEI experiments for CATCRBS. We can obtain the identification of airplanes by SEI, ignoring the effects of different signal modes and emitters. With this, our SEI system can completely replace the identification-decoding module within CATCRBS. To explore the effects of different modal signals transmitted by the same emitters, rm1 and rm2 are generated by the same three Tektronix AWG 5000 series arbitrary waveform generators. These signals are received by the Teledyne LeCroy WaveRunner 9404 oscilloscope and used to build the simulated dataset.

Ads-b, rm1, and rm2 have the same modulation PPM, but with different structures. We utilize their preamble pulses for SEI. The preamble pulses of communication signals without any modulated information are more efficient and less susceptible to deception when identifying emitters. With the same modulation, the proposed MSWTEu can also recognize the signal modes, as shown in later experiments. Wi-Fi signals consist of continuous pulses, and we adopt each pulse as a single signal for SEI.

**Training procedure:** Firstly, RFF features are extracted by MSWTEu from each sample. Non-Wi-Fi signals select preamble pulses, while Wi-Fi signals select one pulse. Secondly, the FTGAN is trained with 3000 unlabeled ads-b MSWTEu (the known mode) and 9000 unlabeled non-adsb MSWTEu (the unknown modes). Thirdly, the Mode 1 SEI classifier is trained with 900 labeled ads-b MSWTEu, as shown in Figure 3. Fourthly, with FTGAN, MSWTEu features of all modes are transferred to match the Mode 1 distribution, and all layers, except for the softmax of the Mode 1 classifier, are used to extract identifying features for all modes. Finally, a clustering model is employed to recognize these 12 emitters based on the identifying features without pre-recognizing signal types and supervised training. If signal mode recognition is required, the clustering model can also be utilized for that purpose.

(**Note:** Although rm1-1∼3 and rm2-1∼3 share the same emitters, the clustering process still divides them into 6 clusters instead of 3. In other words, in our experiments, the final clustering number for SEI is 12. There are two main reasons for this: (1) Different modal signals from the same wireless devices may actually be from different emitters, such as the actual ads-b a/c/s modes. (2) Clustering the same emitter with different modal signals into different clusters can help the SEI system simultaneously identify signal modes and sources. For a convenient description, in the later experiments, adsb-1, adsb-2, adsb-3, wifi-1, wifi-2. wifi-3, rm1-1, rm1-2, rm1-3, rm2-1, rm2-2, and rm2-2 are used as separate transmitter IDs, representing individual emitters, respectively).

However, for a clear classification, 10 labeled samples per cluster are required for label correction. It is worth mentioning that signal labeling should be corrected in cross-domain exploring, which is different from image cross-domain exploration [22,31], while maintaining a similar inherent structure. The cross-domain exploration problem in signal analysis also exists in other areas without any similar inherent properties, such as the conversion of texts to graphs. With proper label correction, the proposed structure of MSWTEu+FTGAN can significantly improve the accuracy of SEI. During the experiments, 300 samples were selected for evaluation by each emitter, with a total of 3600 samples evaluated (4 modes × 3 emitters × 300).

### 4.2. Results and Discussions

**Extractors for identifying devices**: The first experiment involves confirming the unifying feature performance and improving RFF features by MSWTEu. Figure 5 shows that the MSWT features have achieved structure unification for different kinds of signals. Moreover, Figure 5 depicts that the MSWTEu has clear improvement in limiting noises and interferences. The red dashed box in Figure 5 shows that the noises have been well inhibited by MSWTEu. Figure 6 shows the MSWTEu’s clustering performance in identifying 3 emitters under ads-b signals with different SNRs, which is obviously better than MSWT without energy unifying under lower SNRs. Moreover, the performances of our proposed methods, which use SPSO for adaptive parameter determination, are better than ISWTE [13], which is the previous RFF feature extractor we propose to counteract heavy noise. The ISWTE is manually adjusted based on limited decomposing layers. Figure 7 depicts the boxplot performance for identifying 12 emitters. MSWTEu performs far better than MSWT in the different modal signals for SEI, although MSWT is only a bit worse than MSWTEu at identifying three emitters using the same modal signal. Above all, MSWT succeeded at unifying RFF features and enhancing SEI performance, but did not show more robustness in identification across multi-modal signals. Furthermore, MSWTEu achieved better performance in SEI across multi-modal signals based on MSWT.

**Complexity of RFF extractors:** We compared the complexity of our MSWT and MSWTEu with MAp and ISWTE, which are RFF extractors that were introduced in recent references [13,27]. Table 2 illustrates the comparative results, where M represents MSWT and MEu represents MSWTEu. The ‘time’ in Table 2 represents the time taken by these extractors to handle 3000 signals when N=1024. (*N* is the length of multi-modal signals after unification). M is completely based on wavelet functions with a time complexity of O(NlogN), and is slightly smaller than that of MEu. MAp in [27] claims to have a complexity of $O(N^{2})$, but its actual complexity is closer to $O(N^{3})$, which requires more time for feature handling, as shown in Table 2. Although MEu has a higher time complexity than M, the additional time cost is only 0.1 s. Moreover, they share the same space complexity. Compared to ISWTE, MEu demonstrates a faster processing speed despite having the same time complexity. Our MSWTEu has obvious advantages in time and space complexity when compared to many RFF extraction algorithms in signal processing. Although MEu is a little worse than M in complexity, the identifying performances in multi-modal signals or under low SNR have obvious advantages, as shown in Figure 6 and Figure 7. Moreover, the FTGAN output dimensionality is 20, further contributing to the smaller space complexity in the final clustering.

**Open-set signal mode identification by RFF extractors**: To investigate the generality of these RFF extractors across multi-modal signals, we conducted experiments using M, MEu, and MAp to handle the four modal signals. If these features are extracted effectively and the multi-modal features are aligned properly, clustering for signal modes based on these features can also be effectively achieved. Therefore, we evaluated their performances in multi-modal signals by measuring the accuracy of mode identification through clustering. We introduce MAp since [27] proposed an RFF feature extractor similar to ours, which can also achieve emitter identification through clustering. However, all the features of rm1 calculated by MAp are zero, indicating that MAp is unsuitable for rm1 processing. Apart from rm1, the average accuracy of MAp is 50% in regard to recognizing signal modes. In contrast, MSWT achieves an accuracy of 92.6%, and MSWTEu achieves 98.1%. Figure 8 depicts the accuracy comparison concerning the recognition of signal modes across different methods. Figure 9 presents the accuracy confusion matrix for various methods in recognizing signal modes, using the same clustering algorithm. The MSWTEu demonstrates exceptional generalizability to different modal signals and performs consistently well in feature extraction, exhibiting the highest accuracy and most stable performance in signal mode clustering.

**SEI performance comparison by MSWT, MSWTEu, and MSWTEu+FTGAN:**Table 3 illustrates the improvement of each proposed method in identifying emitters. In the experiments, k-means is used for clustering. M uses MSWT+k-means and MEu uses MSWTEu+k-means to classify emitters. MEu+ FTGAN is as depicted in Figure 3 for classification. When the classified number is 3 in Table 3, k-means is only used to identify 3 emitters for the same modal signal. When the classified number is 12 in Table 3, the k-means is utilized to directly distinguish 12 emitters for all modal signals. MSWTEu performs significantly better than MSWT in classifying emitters for the same modal signal but FTGAN does not improve the accuracy for the same modal signal. In the comprehensive recognition of twelve emitters by different modal signals, MSWT only achieves 69% accuracy, and rm2-1∼3 is almost not recognized, which is also shown in Table 3 and Figure 7a. MSWTEu improved by 26% while recognizing rm2-1∼3. However, relying solely on feature extraction, the integrated performance in identifying twelve emitters through different modal signals is still not enough. Although FTGAN is trained unsupervised and no extra labeling work is required, the MSWTEu+FTGAN has nearly 20% improvement compared with MSWT, achieving 86% comprehensive accuracy. In conclusion, MSWTEu offers an improvement in SEI for both the same modal signals and different modal signals when compared to MSWT. This demonstrates the effectiveness of the energy-balancing strategy for the RFF feature extractor. The SEI effect by FTGAN is not improved in the same modal signal compared to MSWTEu since the FTGAN is optimized based on transferring features for multi-modal signals. However, the features transferred by FTGAN show significant improvement in SEI with different modal signals compared to MSWTEu, proving that the multi-modal signal features have been successfully transferred to a matching distribution by the FTGAN. Figure 10 also shows the comparisons between MSWT, MSWTEu, and MSWTEu+FTGAN in terms of recognizing emitters with the same modal signals and multi-modal signals. Figure 10 shows the processing time and accuracy recognition among different methods. In Figure 10a, although the processing time of MSWTEu is longer than that of MSWT, the average accuracy recognition is clearly higher than that of MSWT, confirming that energy balance has an effect on feature extraction but at some cost. Moreover, in Figure 10b, FTGAN shows a clear improvement in recognizing emitters across different signal modes, which confirms that FTGAN successfully matches the multi-modal signal features but with the same time costs.

**The comprehensive SEI performance:**Table 4 depicts the comparison performances of different methods in identifying 12 devices by multi-modal signals. We introduce MAp [27] to compare with our MSWT and MSWTEu. MAp is an RFF extractor and utilizes clustering for identification. MAp is designed for Wi-Fi [27], but it is not suitable for other modal signals. As shown in Table 4, it fails to identify emitters by non-Wi-Fi signals and achieves only 35% average accuracy in SEI by multi-modal signals. DTN is a representative cross-domain method proposed by [22]. In DTN, ads-b is also the known mode. Through training and fine-tuning via multi-modal signal features extracted by MSWTEu, DTN only has 63% accuracy, which is even lower than MSWT (DTN labels have been corrected). The datasets in [22] consist of multi-modal graphs. Although different modal graphs have different modal features, they still share inherent features for identifying objects. Therefore, our experiments show that the method in [22] does not achieve the desired cross-domain SEI in the signal domain. TFL [21] trains a base classifier for one modal signal and then fine-tunes the last few layers to fit other modal signals. Table 4 also suggests that TFL does not perform well in SEI, confronting multi-modal signals from different emitters. In [21], TFL attempts to identify emitters across domains, from ads-b to Wi-Fi. However, the final accuracy in the experiment of [21] is also unsatisfactory, coming across more as an exploratory discussion. Due to the unique wave structure of signals, our work proposes the extraction of a unified and matched feature set for different modal signals, using clustering for emitter identification. Although the clustering needs 10 samples for label correction, the TFL needs more labeled samples for fine-tuning but still has worse results. Moreover, by TFL, a new modal signal requires a new, fine-tuned classifier, which wastes more time and resources compared to the proposed MSWTEu+FTGAN. Figure 11 depicts the comparison curves of the different methods used in identifying 12 devices through multi-modal signals. The red line represents our method, which outperforms all other methods in all modal signals.

In conclusion, identifying wireless devices via cross-modal signals may not be suitable for traditional cross-domain and transfer learning. These methods perform poorly and also require label correction in SEI by cross-modal signals. Therefore, by obtaining a robust RFF distribution across multi-modal signal domains, cross-modal SEI may be better suited for clustering. Moreover, by using a few labels for label correction, we can achieve a higher SEI accuracy. Our MSWTEu and FTGAN unify the RFF features and match the final feature distributions for multi-modal signals, leading to better results in SEI through clustering and significant progress in cross-modal SEI. In Figure 12, we present an overview of the final performance through the confusion matrix for 12 emitters across 4 modal signals. We can see that different modes have been effectively clustered, making it hard for the same mode to be clustered into other modes. Moreover, the comprehensive accuracy of emitter identification aligns with our projections.

## 5. Conclusions 

In this paper, we addressed the challenge of how an SEI system can identify wireless devices across multi-modal signals using only a single RFF extractor and a single SEI classifier. In typical scenarios that encounter multi-modal signals, no prior works have efficiently proposed an SEI system for identifying emitters without the need for different extractors and classifiers for each modal signal. Specifically, the enhanced SEI system with MSWTEu and FTGAN that we propose addresses the above issues. Confronting multi-modal signals, the MSWTEu unifies the features first, and further optimizes feature energy calculation to improve the comprehensive RFF performance. Moreover, with FTGAN and the proposed training process, the MSWTEu of different modal signals can be transferred to a matching distribution, unifying the final identifying multi-modal signal features for SEI and requiring only one unsupervised clustering for the identifying emitters. Experimental results, evaluated on a mixture of simulated, public, and real received datasets, show that MSWTEu, FTGAN, and the proposed training strategy can significantly improve the SEI performance and save resources when dealing with multiple types of signals. Solely using feature tackling, MSWTEu can reach 87% accuracy under the same modal signals in unsupervised clustering for identifying emitters, 75% accuracy under different modal signals in unsupervised clustering for identifying emitters, and 98.1% accuracy in unsupervised clustering for identifying signal modes. The results are far better compared to existing SEI studies that pay attention to feature tackling. With the inclusion of FTGAN and our proposed training approach, the feature performance in cross-modal emitter identification is further enhanced, rising from 75% to 86% in clustering. In conclusion, we believe that further improvements in feature optimization and matching for multi-modal signals and utilizing these improved features through unsupervised clustering to identify emitters have great potential in improving the comprehensive performance in crossing-modal SEI and reducing manual work, such as labeling.

However, there are some limitations in our work: (1) the simple clustering algorithm used for identifying emitters has limited adaptability when confronted with a constantly changing number of emitters; (2) the proposed methods cannot fully resolve the identification problem for new modal signals that lack labels. This issue still requires some manual labeling for correction; (3) the same emitters with different signal modes cannot be clustered into the same class. This can be seen in the boxplot of Figure 7b in rm1 and rm2, as the rm1 and rm2 modes share the same emitters. Future work on SEI cross-multiple modal signals should be further improved (1) to research a proper dynamic clustering algorithm to replace the simple clustering in our SEI system and solve limitation 1; (2) to research a feature extraction and matching method that is more adaptable and robust, to reduce limitation 2 and resolve limitation (3). 

## Figures and Tables

**Figure 1 sensors-23-08576-f001:**
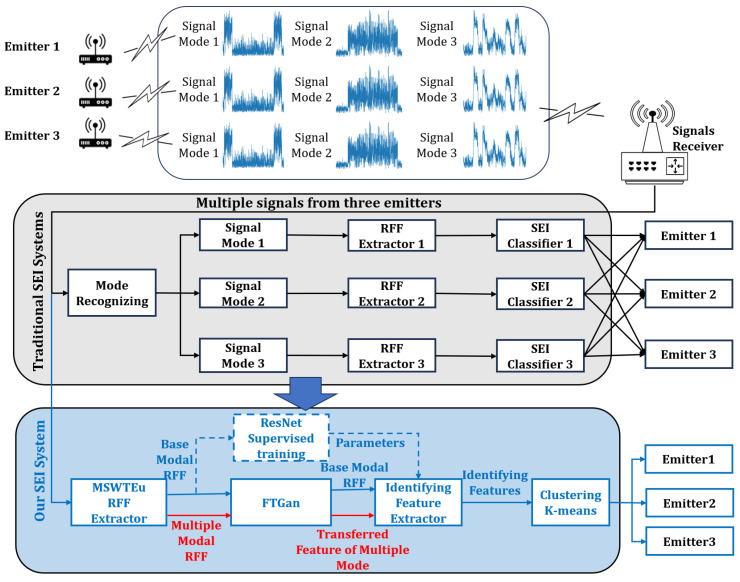
The process of a traditional SEI system for multi-modal signals and our SEI system for multi-modal signals. Our system aims to unify the extractor and classifier, which improves the efficiency of the system. With this improvement, the system decreases labeling work.

**Figure 2 sensors-23-08576-f002:**
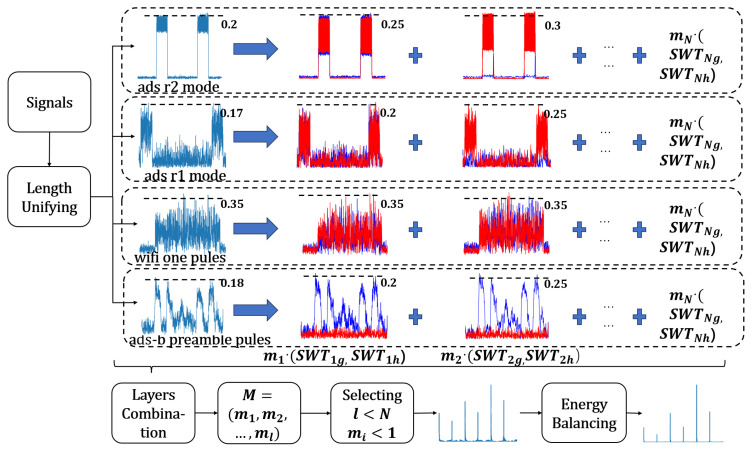
The MSWTEu process. The first column waveform is the time-domain wave; in the second to third columns, the blue wave represents $SWT_{ig}$, and the red wave represents $SWT_{ih}$.

**Figure 3 sensors-23-08576-f003:**
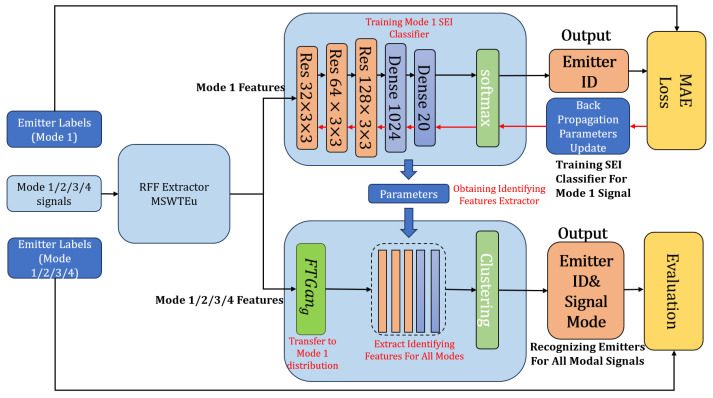
The emitter label identification procedure for a single mode (Mode 1) and four different modes (Mode 1/2/3/4). The top right part is the SEI classifier trained for Mode 1 under supervision, which outputs the emitter ID. The bottom right is our multi-modal SEI system, which recognizes the emitter ID for all modal signals. Our multi-modal SEI system uses FTGAN to transfer different mode features to Mode 1 distribution and shares parameters with the Mode 1 SEI classifier.

**Figure 4 sensors-23-08576-f004:**
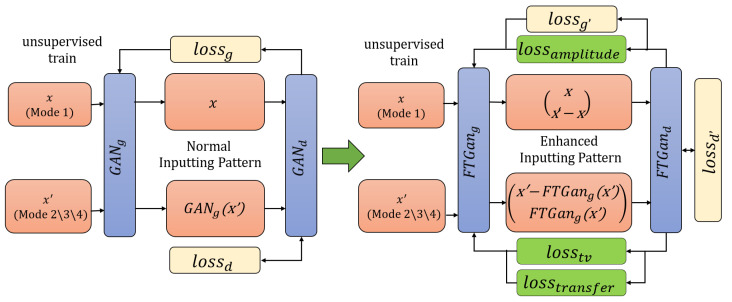
The architectural comparison between normal GAN and FTGAN.

**Figure 5 sensors-23-08576-f005:**
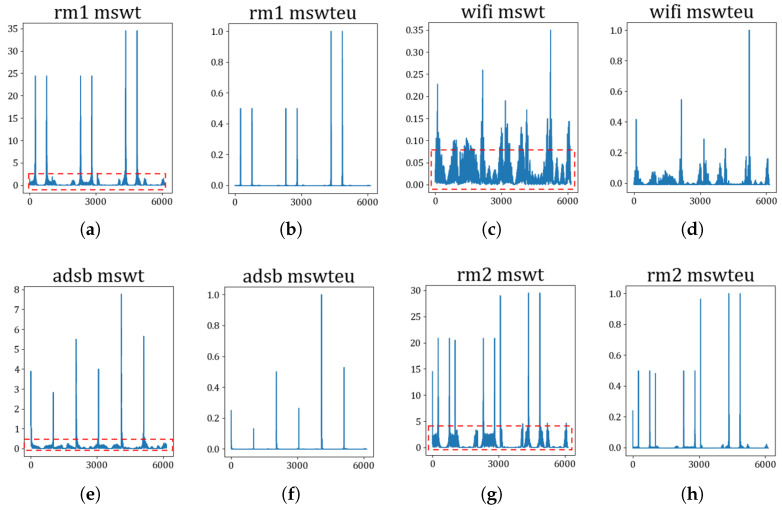
The waveform feature comparison between MSWT and MSWTEu; (**a**,**c**,**e**,**g**) are the MSWT waveform features of rm1, wifi, ads-b, and rm2; (**b**,**d**,**f**,**h**) are the MSWTEu waveform features of rm1, wifi, ads-b, and rm2; the red boxes are noises not decreased by the MSWT, which do not exist in the MSWTEu waveform features; the horizontal axis displays the frequency and the vertical axis displays the amplitude.

**Figure 6 sensors-23-08576-f006:**
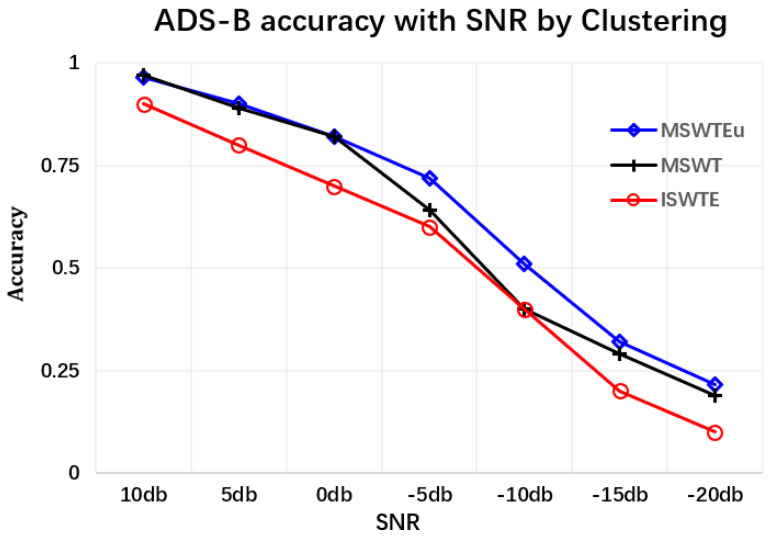
The SEI accuracy comparison between MSWTEu, MSWTE, and ISWTE in identifying emitters by ads-b signals under different SNRs.

**Figure 7 sensors-23-08576-f007:**
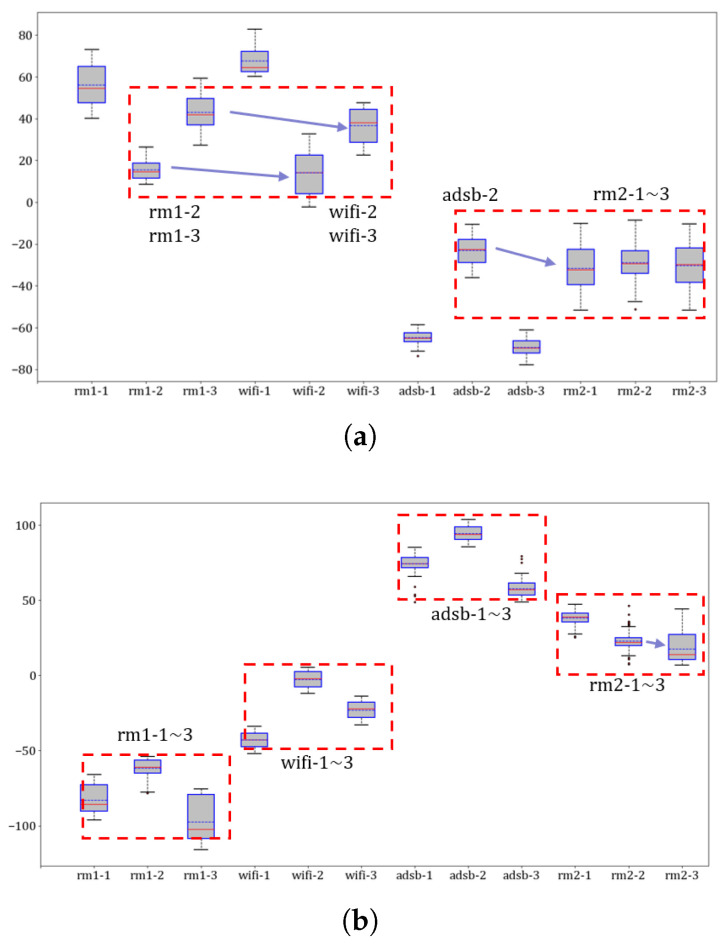
Boxplot representation of MSWT (**a**) and MSWTEu (**b**) for 12 devices; the values of the same red boxes show a similarity which crosses different modes in (**a**) but not crosses different modes in (**b**); the horizontal axis represents the signal mode while the vertical axis indicates the value of each feature. For visualization purposes, these features have been reduced to 1 dimension using T-distributed stochastic neighbor embedding (TSNE).

**Figure 8 sensors-23-08576-f008:**
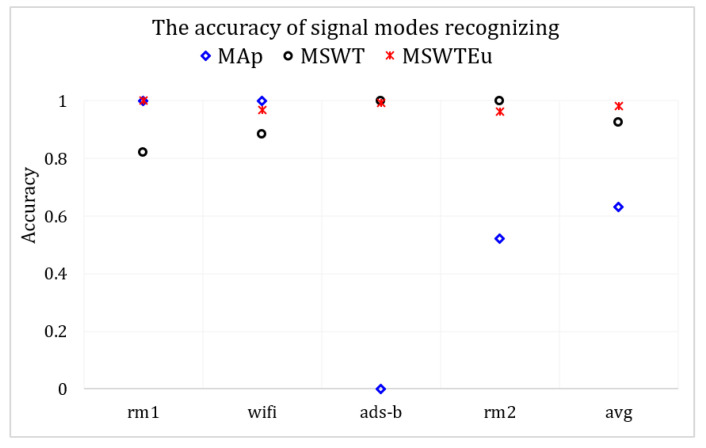
The accuracy comparison for various methods in recognizing signal modes.

**Figure 9 sensors-23-08576-f009:**
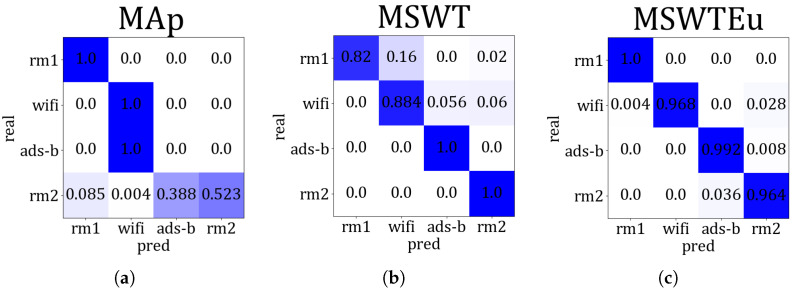
The accuracy confusion matrix for various methods in recognizing signal modes, (**a**) is MAp, (**b**) is MSWT, (**c**) is MSWTEu, and the darker color represents the higher accuracy.

**Figure 10 sensors-23-08576-f010:**
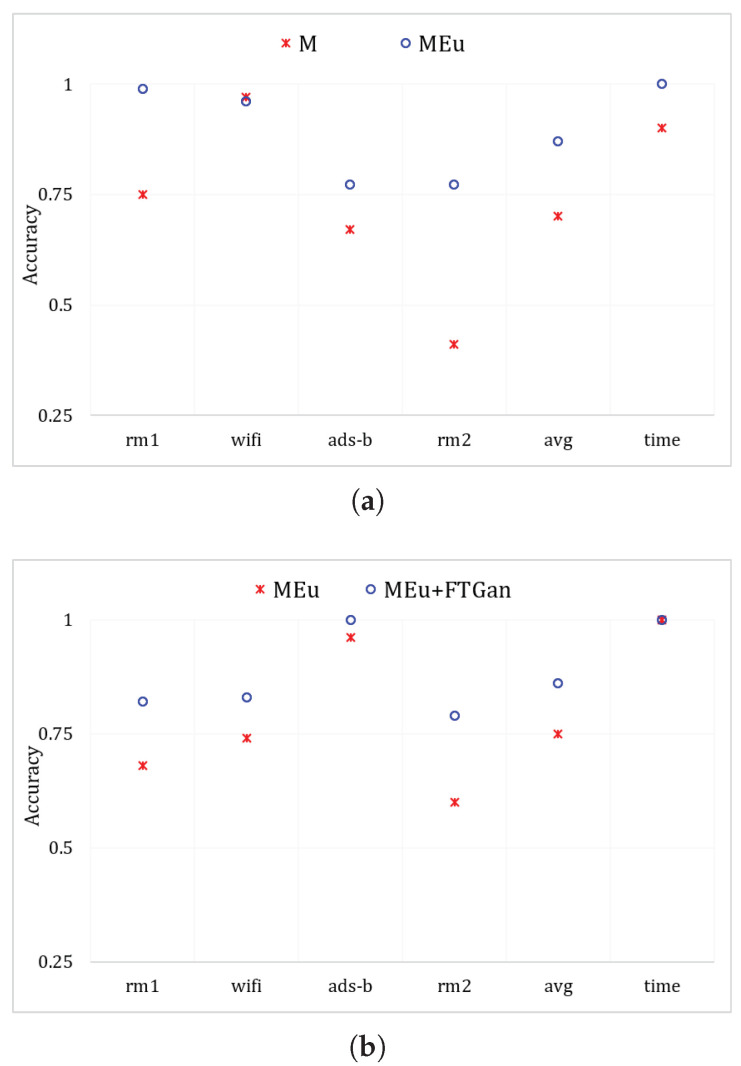
MSWT and MSWTEu performances in recognizing 3 emitters for the same mode, (**a**) and MSWTEu and MSWTEu+FTGAN performances in recognizing 12 emitters for multiple modes (**b**).

**Figure 11 sensors-23-08576-f011:**
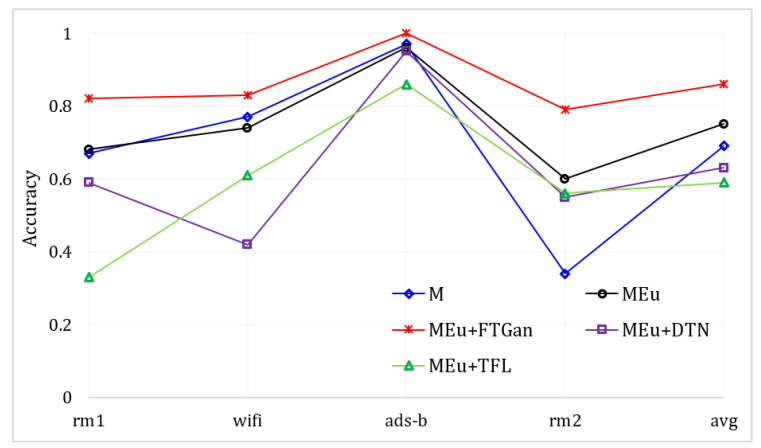
The SEI accuracy through multi-modal signals based on different methods.

**Figure 12 sensors-23-08576-f012:**
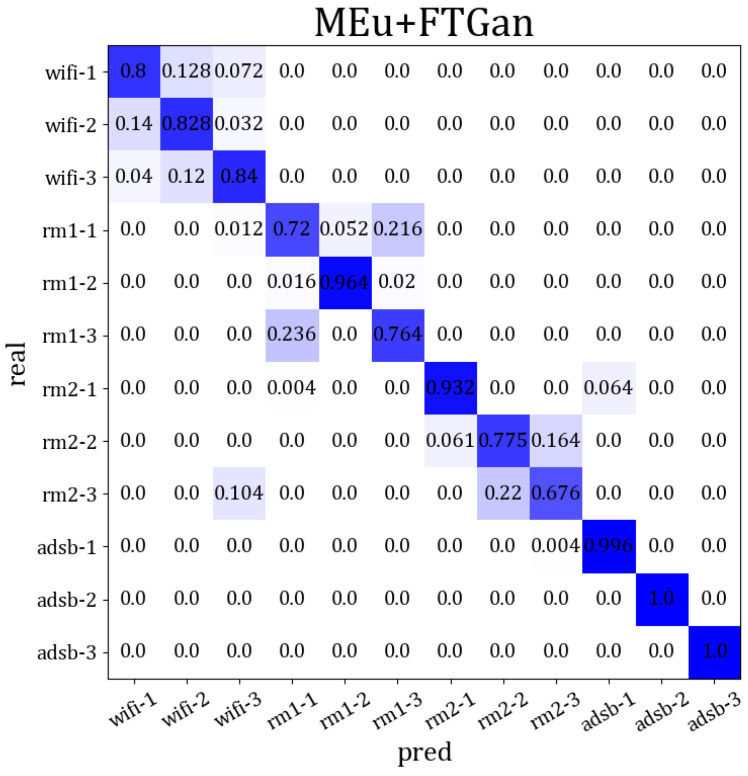
The comprehensive SEI accuracy confusion matrix through multi-modal signals based on MSWTEu+FTGAN, and the darker color represents the higher accuracy.

**Table 1 sensors-23-08576-t001:** Details of experimental signals.

Signal Mode	Transmitter ID	Details	Training
ads-b (ads-b s mode)	adsb-1 adsb-2 adsb-3	Aircraft communication signals, collected from real aircraft by a real-time spectrum analyzer over a 12-month period; The preamble pulses are for SEI, including 4 pulses for each signal; $f_{c}=1090$ MHz, $f_{s}=150$ MHz, PPM Modulation, SNR = 10 db.	3000 unlabeled samples in FTGAN training as the known mode; 900 labeled samples in classifier training; 900 labeled samples in classifier evaluation.
Wi-Fi	wifi-1 wifi-2 wifi-3	Public dataset [18] part 3 raw data; wifi-1∼3 are from 3 antennas of the same device; One transmitted pulse is for SEI per one signal; IEEE 802.11a/g, 2.4 GHz, 20 MS/s, BPSK Modulation, SNR = 5 db.	9000 unlabeled samples (3000 per 1 mode) in FTGAN training as the unknown modes; 2700 labeled samples (300 per 1 mode per 1 emitter) (Wi-Fi, rm1, rm2) in classifier evaluation.
rm1 and rm2 (ads-b a/c mode)	rm1-1 rm1-2 rm1-3 rm2-1 rm2-2 rm2-3	Aircraft communication replying signals, generated by three waveform generators and collected by an oscilloscope; rm1-1∼3 and rm2-1∼3 share the same generators; The preamble pulses are for SEI, including 2 pulses for each signal; $f_{c}=1030$ MHz, $f_{s}=1030$ MHz, PPM modulation, SNR = 10 db.	

**Table 2 sensors-23-08576-t002:** The complexity comparison between different RFF extractors.

Methods	Time Complexity	Time	Dimension
M	O(NlogN)	0.9 s	6N
MEu	$O(N^{2})$	1 s	6N
ISWTE	$O(N^{2})$	10 s	6N
MAp	≈$O(N^{3})$	1 d	4
MEu+FTGAN	–	–	20

**Table 3 sensors-23-08576-t003:** SEI accuracy of MSWT, MSWTEu, and MSWTEu+FTGAN confronting one modal signal and multi-modal signals.

Classify Number	3 Emitters under the Same Modal Signals			12 Emitters under 4 Modal Signals		
Device ID/Method	M	MEu	MEu+FTGAN	M	MEu	MEu+FTGAN
rm1-1∼3	0.75	0.99	0.865	0.67	0.68	0.82
wifi-1∼3	0.97	0.96	0.815	0.77	0.74	0.83
adsb-1∼3	0.67	0.772	0.99	0.97	0.96	1
rm2-1∼3	0.41	0.772	0.78	0.34	0.6	0.79
Avg 3	0.70	0.87	0.865	/	/	/
Avg 12	/	/	/	0.69	0.75	0.86

**Table 4 sensors-23-08576-t004:** SEI accuracies of different methods confronting multi-modal signals.

Method	rm1-1∼3	wifi-1∼3	adsb-1∼3	rm2-1∼3	Avg 12
M	0.67	0.77	0.97	0.34	0.68
MEu	0.68	0.74	0.964	0.6	0.75
MAp	0.29	0.42	0.34	0.36	0.35
MEu+DTN	0.59	0.42	0.95	0.55	0.63
MEu+TFL	0.33	0.61	0.86	0.56	0.59
MEu+FTGAN	0.82	0.83	1.0	0.79	0.86

## Data Availability

Not applicable.

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
