# Peer review of "A Specific Emitter Identification System Design for Crossing Signal Modes in the Air Traffic Control Radar Beacon System and Wireless Devices"

_sensors, 2023, doi:10.3390/s23208576_

Round 1

Reviewer 1 Report

The paper presents an enhanced Specific Emitter Identification (SEI) system utilizing a universal radio frequency fingerprint (RFF) extractor and a feature transfer Generative Adversarial Network (FTGan) to recognize emitters across various modal signals with a single SEI classifier. I have the following comments:

 -        For the reader, understanding the abbreviations in the article's title is difficult.

-        I recommend using a consistent style for writing abbreviations in the article. The initial letters of the abbreviated words should be capitalized, e.g., Specific Emitter Identification (SEI).

-        Please provide an explanation of the abbreviation DARPA.

-        All figures should be stretched to fit the entire page. I recommend the author to enlarge the captions in the images.

-        The authors claim that the work was divided into three stages. Would it be possible to somehow indicate these stages in Figure 2?

-        Starting from line 154, the MSWTEu calculation process is described. Each point begins with an abbreviation. Would it be possible to write the full title instead of these abbreviations?

-        The equations are part of the sentence, so they should be followed by punctuation marks, a comma if the sentence continues, or a period if the sentence ends.

-        Please clarify what Dense20 means on line 404.

-        Please add axis labels in the graphs for Figure 5 and Figure 7.

-        If I understood correctly, in the conclusion of the article, you argue that manual optimization of the system is necessary if emitters or the environment changes. Could you comment on this? What adjustments would be necessary?"

Reviewer 2 Report

Paper is well written. Methods are sound and conclusion well supported by the results.

Some concerns are raised as follows: 

1. The authors need to refine the abstract in a more précised way by identify the main purpose, research question, hypothesis, methodology, results with values, and conclusions of your research. 

2. Give a summary of your related work and define your problem statement clearly as a separate section, also add a para in introduction paper organization

3. Proposed model can be explained with a clear architectural diagram involving all your components and with more clarity in the diagram 

4. The study's fundamental structure has to be explained with the help of an algorithm

5. How have the outcomes been ensured in light of the major uncertainties? 

6. Strengthen the results section by adding tables and graphs

The conclusion has to be revised to incorporate the following advice: - 

- Highlight your analysis and just present the most important takeaways from the full paper. 

- Mention the advantages. 

- In the final sentence of this section, mention the inference. 

- Make sure the topic of the Conclusion differs from what is discussed in the abstract. 

- include the future work with multidimension work

minor corrections

Reviewer 3 Report

The paper "A Specific Emitter Identification System Design For Crossing

Signal Modes in ATCRBS and Wireless Devices" the authors are proposing an improved system to solve the problem of identifying wireless devices. The topic is interesting and the authors present well the problem of identifying wireless devices using modal signals. But there are some small details to be adjusted:

1-Figure 1 must be enlarged so that you can check the details in the block diagrams. The same problem occurs in Figure 2.

2- Equation 4 can be divided into two equations, making it clearer and more evident in the text.

3-A space between equations 8a, 8b and 9a and 9b. Instead of 8a, 8b, 9a and 9b it could be 8, 9, 10, 11 and 12. The same must be done throughout the text with the other equations.

4-I didn't understand equation 12? I believe it does not need to be in the text.

5-Figures 3, 4, 5, 6 and 7 allow you to see the information in the figure in detail. I suggest expanding or increasing the strength of the information.

6-Just a suggestion, future work should be in complementary material or after the speeches. In conclusion, only the final observations of the work.

The paper needs an English revision.
